# Sport Psychological Skill Factors and Scale Development for Taekwondo Athletes

**DOI:** 10.3390/ijerph19063433

**Published:** 2022-03-14

**Authors:** Jung-Hoon Nam, Eung-Joon Kim, Eun-Hyung Cho

**Affiliations:** 1Department of Sports Healthcare, Catholic Kwandong University, Gangneung 25601, Korea; n7j7h782@gmail.com; 2Department of Physical Education, Korea National Sport University, Seoul 05541, Korea; ejkim@knsu.ac.kr; 3Department of Sport Science, Korea Institute of Sport Science, Seoul 01794, Korea

**Keywords:** taekwondo athletes, sport psychological skill, item response theory, integrated validity

## Abstract

The purpose of this study was to identify the sport psychological skills of Taekwondo athletes and to develop a scale measuring such skills. We collected preliminary data using an open-ended online survey targeting Taekwondo athletes from nine countries (South Korea, China, Malaysia, United States, Spain, France, Brazil, United Kingdom, and Taiwan) who participated in international competitions between 2019 and 2020. We extracted participants’ sport psychological skills from 75 survey responses, guided by expert meetings and a thorough literature review. We verified our Taekwondo psychological skill scale’s construct validity using 840 survey responses. We utilized V coefficients, parallel analysis, an exploratory structural equation model, maximum likelihood, confirmatory factor analysis, and multi-group confirmatory factor analysis for data analysis. We identified six core sport psychological skills: “goal setting,” “confidence,” “imagery,” “self-talk,” “fighting spirit,” and “concentration.” Our final measure, which demonstrated evidence of reliability and validity, comprises 18 items spanning 6 factors, with each item rated on a 3-point Likert scale.

## 1. Introduction

Athletes’ performance relies as much on mental strength as on physical strength and athletic skills. This means that psychological skills are crucial for optimal athletic performance [1]. According to a study by Orlick and Partington [2], the psychological readiness of Canadian national athletes competing in the 1984 Los Angeles Olympics majorly influenced the number of medals won. In other words, their psychological training process—including goal setting, psychological preparation for the actual competition, and coping strategies for uncertain situations—was highly effective. In agreement with this, studies on athletes’ psychological characteristics [3,4,5,6] show that high-performing athletes have higher confidence and lower anxiety levels than less successful athletes. Researchers also report that athletes with clear goals actively cope with anxiety and have excellent concentration, mental preparation, and ability to immerse themselves in the competition as well as to relax. Accordingly, it is becoming increasingly important for athletes to cultivate psychological coping abilities in games or training situations to maximize their performance.

The readiness to cope with various emotional or psychological problems in sports is referred to as sport psychological skills [7,8,9]. Sport psychological skills are techniques and strategies for achieving optimal performance by regulating negative psychological factors that interfere with the competition, such as anxiety, fear, and frustration [10]. Therefore, sport psychological skills are considered essential for stable athletic performance [11].

In sport classification, closed-skill sports and open-skill sports are divided according to the connection status of the mind–body feedback, and more than 80% of the performance in closed-skill sports is determined by psychological factors [12,13]. Accordingly, sport psychological skills have long been used in closed-loop skill sports such as shooting and archery, but their use is gradually expanding to open-loop skill sports such as skating and gymnastics [12,13].

Targets are also expanding, as even athletes who are not experiencing psychological problems can still apply these skills. In accordance with this trend, sport psychological skills are constantly changing and developing. It is possible to define known sport psychological skills as imagery, relaxation, goal setting, and concentration, so that they reflect the characteristics of specific sports and the diversity of the training and competitive conditions.

Taekwondo has 209 member countries, being a worldwide sport adopted in 16 single international competitions and 15 countries’ comprehensive international competitions [14]. Taekwondo is a martial art where victory or defeat in a match is determined by striking one’s opponent. Athletes face a great psychological burden during competitions; thus, not only strong physical strength and skills but also psychological management of fear and anxiety are important in Taekwondo [15]. There have been numerous studies of Taekwondo athletes’ psychological skills [15,16,17,18,19], revealing that relaxation, self-talk, and imagery are especially closely related to the athletic performance. However, despite uncovering the relationship between performance and sport psychological skills, these studies also have limitations, in that they did not determine the mechanisms or created measures of sport psychological skills specifically reflecting Taekwondo. Therefore, we argue that it is necessary to perform a structural exploration of Taekwondo athletes’ sport psychological skills and to develop a scale to measure them. This would lead to a more comprehensive understanding of Taekwondo athletes’ performance, including ways to maximize it through skills training.

Scale development studies in the social sciences and sports are largely divided into two categories: (1) those that use the ternary analysis of content validity, construct validity, and criterion-related validity along with the classical test theory [20] and (2) those that develop scales based on the Rasch model of item response theory and integrated validity as a criterion [21,22,23]. However, the ternary analysis method may distort the factor structure due to researchers’ subjectivity when creating items and extracting factors from the theoretical model [24]. Additionally, there are obstacles impairing the scale’s validity and reliability, including the inability to use various objective indicators, which reduces the replicability of the theoretical model captured by the scale [25,26,27,28,29,30]. To address these limitations, researchers have recently employed Benson’s [31] construct validity approach as an alternative scale development method that utilizes the item response theory and Messick’s [32] integrated validity concept as a criterion, which was adopted as a standard in the Standards for Educational and Psychological Testing suggested by the American Educational Research, American Psychological Association, and National Council on Measurement in Education [33]. Based on the integrated validity concept, Benson’s [31] construct validity approach considers not only the test subject’s validity and reliability but also the reliability and difficulty of the items themselves. Therefore, it has the advantage of being able to examine scales objectively, regardless of individuals’ characteristics [29]. In the process of developing a scale, the method also analyzes the constructs to be measured, using the common factor model and structural equation for each substantive, structural, and external domain; further, it utilizes all data from theoretical and practical perspectives to improve the scale’s psychometric properties [26,27,31]. Therefore, to develop a scale by empirically reflecting the given theoretical model and individuals’ psychological characteristics, the construct validity approach appears appropriate [31].

The purpose of this study was to explore Taekwondo athletes’ sport psychological skills by applying the integrated validity concept and construct validity approach, ultimately developing a scale measuring these skills. This study’s results may provide a strategic reference for identifying Taekwondo athletes’ psychological abilities and thereby improving their athletic performance.

## 2. Methods

### 2.1. Participants and Data Collection

Research participants included an expert group and an athlete group. The expert group enabled us to extract the sport psychological factors and to test the scale items’ content validity. Accordingly, we recruited sports psychology experts with professional knowledge or experience in Taekwondo, using the purposive sampling method, as shown in Table 1.

As shown in Table 2, the expert group consisted of three sports psychology professors, three national Taekwondo coaches with more than 10 years of experience, and three Taekwondo athletes with more than 5 years of national team experience.

Meanwhile, we conducted an online self-report survey with individuals from nine countries (South Korea, China, Malaysia, US, Spain, France, Brazil, UK, and Taiwan) who participated in international Taekwondo competitions between 2019 and 2020 to analyze the construct validity of our Taekwondo sport psychological skill scale.

The surveys took the form of either an open-ended questionnaire or Likert-scale items. We collected 75 total responses to the open-ended questionnaire and 882 total responses to the Likert-scale questionnaire. However, we eliminated 42 responses due to careless responding, which yielded a total of 840 responses for the analysis. Among these responses, we randomly divided those to the Likert-scale questionnaire in two groups (Group A/Group B) for analysis.

### 2.2. Data Analysis

In this study, we analyzed the substantive domain, structural domain, and external domain according to Benson’s [31] construct validity procedure, while also utilizing the Rasch model based on the concept of integrated validity in Table 3.

In the substantive domain, we generated items based on the conceptual structure of Taekwondo athletes’ sport psychological skills, and then experts evaluated the items’ appropriateness through content validity verification using Aiken’s [34] V factor. Additionally, using WINSTEPS 3.65 [35], we conducted unidimensionality verification and evaluated the appropriateness and relevance of the items’ response categories. In the structural domain, we conducted parallel analysis and used the exploratory structural equation model to explore the factor structure of the sport psychological skill scale using Mplus 7.4. To verify the suitability of the sport psychological skill factor structure yielded by the exploratory structural equation model, we conducted a confirmatory factor analysis. Lastly, in the external domain, we conducted a latent mean average analysis based on gender to determine the sport psychological skill scale’s validity.

## 3. Results 

### 3.1. Substantive Domain

#### 3.1.1. Extraction of Taekwondo Athletes’ Sport Psychological Skills

To identify Taekwondo athletes’ sport psychological skills, we conducted a thorough literature review [15,16,17,18,19] and administered an open-ended questionnaire to the partcipants. We extracted relaxation, self-talk, emotional regulation, imagery, goal setting, routine, self-confidence, and concentration as sport psychological skills based on the literature review. We further investigated these skills by asking the open-ended question “What kind of psychological control methods or techniques are applied during training and competition?” Responses to the open-ended questionnaire—discussed by experts using the domain-referenced test and clarity theoretical applications of Devellis [36] and Hively [37]—revealed the sport psychological skills of confidence, fighting spirit, will, and routine. Ultimately, the literature review and expert discussion of the open-ended questionnaire responses yielded 10 potential sport psychological skills used by Taekwondo athletes: relaxation, self-talk, emotion regulation, imagery, goal setting, routine, confidence, concentration, fighting spirit, and will in Table 4.

#### 3.1.2. Item Generation

We generated items through discussion among experts about the sports psychology skills of Taekwondo athletes identified through the classification process. Following the method suggested by Crocker and Algina [38], experts deliberated on five items for each factor considering the content, grammar, and meaning conveyed. Then, we applied a 5-point Likert scale to each item, finding the most common response category. To assess content validity, we utilized Aiken’s [34] V coefficient and the binomial probability distribution for each item in Table 5.

Consequently, 12 (#4, #7, #13, #14, #18, #21, #25, #30, #36, #43, #46, #48) of the original 50 items had V coefficients near 0, with significance levels of 0.05 or greater. This means that these items were not suitable for measuring the respective sport psychological skills. Therefore, we eliminated these items and subjected the remaining 38 items to further analysis.

#### 3.1.3. Verification of Unidimensionality

Before examining the construct validity of the factor structure, we used principal components analysis to verify the construct’s unidimensionality, which is a prerequisite for applying the Rasch model. As shown in Table 6, the explained variance was 42.2%, so the criterion of 20% or higher observed variance, which is the condition for unidimensionality, was satisfied [39].

#### 3.1.4. Appropriateness of the Response Range

To verify the appropriateness of the 5-point Likert scale, we used the Rasch model’s rating scale analysis. The adequacy of the scale is based on the probability curve for each category, guided by the following questions. First, are there at least 10 observations (count) for each category? Second, are the frequencies and proportions of the categories evenly distributed? Third, as the categories increase, does the average measure of the categories increase? Fourth, do the standardized infit and outfit values of each category satisfy the fit criteria (7.50–1.30; [40]) Fifth, does the step calibration change from at least 1.4 to less than 5.0?

As shown in Table 7, the observed values, frequency (%), average measurement (AM), and infit and outfit values for each category, were within the acceptable range. However, step calibration values using the 2-point scale and 3-point scale, 4-point scale, and 5-point scale were less than 1.4. This means that they did not meet the eligibility criteria, so the 5-point Likert scale was inappropriate to measure these sports psychological skills.

Accordingly, we conducted step calibration using the 3-point and 4-point Likert scales. As shown in Table 8, the 3-point Likert (11223) scale satisfied the fit criteria. Therefore, we concluded that the 3-point Likert scale was suitable for measuring Taekwondo athletes’ sport psychological skills.

#### 3.1.5. Item Relevance

To evaluate the scale items’ relevance, we applied the Rasch model’s rating scale [41]. Item relevance refers to the discriminatory level of each item. We evaluated the items’ suitability using the MNSQ index and the point-biserial correlation (PBC) coefficient in Table 9.

The value of the MNSQ index was between 0.75 and 1.30, the standard suggested by McNamara [40], and the point-biserial correlation coefficient was more than 0.30, the standard suggested by Wolfe and Smith [28]. The MNSQ index for 12 of the 38 items was less than 0.75 or greater than 1.30. This result means that these 12 items were not suitable for measuring Taekwondo athletes’ sport psychological skills; therefore, we eliminated them from the scale.

### 3.2. Structural Domain

#### 3.2.1. Parallel Analysis

To investigate the scale’s factor structure, we used Mplus 7.4 to perform exploratory structural equation modeling and confirmatory factor analysis with the 26 items determined by the substantive domain. First, we conducted parallel analysis to determine the number of factors. As shown in Table 10, the real-data eigenvalue for Factor 6 was lower than 95% of random eigenvalues; thus, we concluded that using five factors to measure Taekwondo athletes’ sport psychological skills was appropriate [42].

#### 3.2.2. Exploratory Structural Equation Model

With parallel analysis yielding a five-factor solution, we extracted factor structure utilizing a model fit index within the range of ±2 and the interpretability of the factor structure [36].

As shown in Table 11, the RMSEA for Factor 4 was less than 0.08; therefore, we identified a factor model with ideal interpretability [43,44] as the factor structure for our scale by comparing factor coefficients for Factors 4 through 7. 

We conducted the ESEM analysis based on the six factors because the six-factor structure was the most suitable theoretical model of Taekwondo athletes’ sport psychological skills (Table 12). In evaluating the six-factor structure through ESEM analysis, multidimensionality was statistically significant and greater than 0.20 for two or more factors. Therefore, through repeated estimation, we eliminated items demonstrating multidimensionality [45] and those hindering practical interpretation due to the factor’s item organization being completely different from the that of the theoretical model [36]. Through this process, we removed 8 total items, with the final scale comprising 18 items spanning 6 factors

#### 3.2.3. Maximum Likelihood Confirmatory Factor Analysis

We conducted a confirmatory factor analysis (CFA) with maximum likelihood (ML) estimation to verify the fit of the factor structure. As shown in Table 13, the RMSEA was 0.053, and the TLI was 0.910, indicating that the model satisfied the criteria for acceptable fit.

Further, the analysis of the factor structure’s convergent validity and discriminant validity revealed that the AVE values and conceptual reliability values of each factor exceeded 0.50 and 0.70, respectively. These results indicated that our sport psychological skills scale for Taekwondo athletes demonstrated strong evidence of convergent validity in Table 14.

To evaluate discriminant validity, we compared the squared value of the correlation (0.487) between Factor B and Factor E (i.e., the highest inter-factor correlation) and the AVE values of Factors B and E. Because the latter exceeded the former, we concluded that there was strong evidence of the scale’s discriminant validity.

As mentioned previously, exploratory structural equation modeling and confirmatory factor analysis yielded a scale with 18 items spanning 6 factors. We named the factors based on the constituent items’ content (Table 12). Thus, we named the first factor “goal setting,” the second factor “confidence,” the third factor “imagery,” the fourth factor “self-talk,” the fifth factor “fighting spirit,” and the sixth factor “concentration.”

#### 3.2.4. Reliability and Difficulty by Factor

To assess the scale’s reliability, we examined both the responses and the items. As shown in Table 15, the separation index and response reliability values for the factors exceeded 2.00 and 0.80, respectively. Further, the infit (internal fit) and outfit (external fit) fell between 0.75 and 1.30. These results indicated strong reliability for the items and the responses. Overall, we concluded that our scale can accurately measure Taekwondo athletes’ sport psychological skills.

### 3.3. External Domain

#### Multi-Group Confirmatory Factor Analysis

To validate the scale, we conducted a measurement equivalence verification using multi-group confirmatory factor analysis (MCFA) by gender. As a result, the ∆X2 from the unconstrained model to the model with similarly constrained factor coefficients was nonsignificant at the 0.05 level (*p* = 0.122). This indicated that there were no gender differences with respect to the scale’s validity, further supporting the scale’s appropriateness for measuring Taekwondo athletes’ sport psychological skills in Table 16.

## 4. Discussion

The purpose of the study was to explore the sport psychological skills exhibited by Taekwondo athletes and to develop a scale measuring these skills. We conducted a literature review and administered open-ended surveys, subsequently relying on expert deliberations using the domain-based method to classify Taekwondo athletes’ sport psychological skills. The expert meetings yielded the skills of relaxation, self-talk, emotion regulation, imagery, goal setting, routine, confidence, concentration, fighting spirit, and will. Next, experts generated items pertaining to each skill and examined their accuracy, grammar, and match with the purpose of each skill. They eliminated negative questions and double-choice questions, using the method suggested by Crocker and Algina [38]. To reduce the subjective biases introduced in the process of producing the questions [46], we verified the items’ content validity using Aiken’s [34] V coefficient.

We verified the construct validity of the scale through the integrated validity approach and the Rasch model. Then, we used parallel analysis and exploratory structural equation modeling to explore the factor structure. The exploratory factor analysis model has the advantage of being able to identify the optimal factor structure because it is possible to check the significance and effect size of the items’ factor coefficients and it is desirable to ascertain the interpretability of the factor structure. Therefore, we evaluated the items’ appropriateness based on the statistical significance of the factor coefficients, as proposed by Jennrich [47]. We refined the factor structure by eliminating items with low discriminatory power (significant cross-loadings with magnitudes of 0.20 or greater). This process ultimately yielded 6 factors and 18 total items, and ML CFA analysis confirmed the suitability of this structure. Finally, we confirmed the scale’s concentration validity, discriminant validity, and item and response reliability and we demonstrated through MCFA that the scale is suitable for measuring Taekwondo athletes’ sport psychological skills, irrespective of gender.

We found Taekwondo athletes’ primary sport psychological skills to be goal setting, confidence, imagery, self-talk, fighting spirit, and concentration. This finding expands on previous studies [15,16,17,18,19] identifying the key skills of self-talk, relaxation, and imagery. With respect to the newly identified skills, unlike in team sports, Taekwondo determines victory or defeat through a one-on-one match. To win a match, a strong mind, confidence, and fighting spirit are essential. Taekwondo coaches and athletes who attended our expert meetings also mentioned that having strong offensive power is a prerequisite for winning Taekwondo matches. Additionally, a strong concentration is required to react quickly and defend against opponents’ attacks. Therefore, the newly discovered skills of confidence, fighting spirit, and concentration can be regarded as accurately reflecting competitive Taekwondo scenarios.

On the other hand, Taekwondo athletes also reported self-talk, which is typically found in athletes of static sports such as golf, as a psychological technique they use. Self-talk involves turning negative emotions or psychological situations into positive ones. Therefore, although sports psychology skills depend on the characteristics of each sport, self-talk appears to be relatively universal.

## 5. Conclusions 

From our exploration and development of a scale measuring Taekwondo athletes’ sport psychological skills, we can draw several conclusions. First, the six major sport psychological skills demonstrated by Taekwondo athletes in training and competition situations include goal setting, confidence, imagery, self-talk, fighting spirit, and concentration. Second, we found evidence supporting the psychometric soundness of a sport psychological skill scale for Taekwondo athletes consisting of 18 items spanning 6 factors, with each item rated on a 3-point Likert scale. Third, our multi-group confirmatory factor analysis verified the scale’s validity regardless of gender.

### Suggestions for Future Research

Although this study investigated the sport psychological skills of Taekwondo athletes, it was not able to determine whether these skills are effective emotional regulation strategies for improving performance. Moreover, the study’s investigation of whether the integrated validity approach and item response theory effectively controlled the transformation of the scale’s factor structure, which is considered a problem with classical test theory in the scale development process, was limited. Therefore, we make several suggestions for future research.

First, studies have shown that sports psychology skills have a positive effect on Taekwondo athletes’ performance [2,3,4,5,6]. However, these studies have limitations in generalizing the contribution of sport psychological skills to Taekwondo based on their role in other sports. Therefore, we recommend follow-up studies investigating the relationship between psychological factors related to performance and sport psychological skills using the scale developed in this study.

Second, there has been insufficient research on improving sport psychological skills themselves and, thereby, Taekwondo athletes’ performance. Therefore, we recommend practical studies aimed at improving the level of sport psychological skills, including the skills of goal setting, confidence, imagery, self-talk, fighting spirit, and concentration identified in this study.

Finally, in general, there are reported differences in performance depending on athletes’ sport psychological skill levels. However, research has not identified such differences with respect to Taekwondo. Therefore, we recommend that future studies use our scale to identify the value and role of sport psychological skills in Taekwondo events according to the performance level.

## Figures and Tables

**Table 1 ijerph-19-03433-t001:** General characteristics of the expert group.

Group		Gender	Number
Experts	Professor/doctor In sport psychology	Male	2
		Female	1
	Coach	Male	2
		Female	1
	Athlete	Male	1
		Female	2

**Table 2 ijerph-19-03433-t002:** General Characteristics of the Survey Participants.

Domain				N		%		
Open-ended survey		Coach	Male	18		60.0		
			Female	12		40.0		
		Athlete	Male	25		55.5		
			Female	20		44.5		
Group A	character		N	Group B	character			N
	sex	Male	277		sex		Male	281
		Female	133				Female	149
	career	Less than 4 years	35		career		Less than 4 years	42
		5 to 7 years	123				5 to 7 years	144
		8 to 10 years	166				8 to 7 years	157
		More than 11 years	86				More than 11 years	87

**Table 3 ijerph-19-03433-t003:** Composition of the questionnaire.

Domain	Analysis	Data Source
Substantive	Literature review	Open-ended Survey
	Inductive categorization/item development according to theoretical model/content validity verification	
	Unidimensionality verification Response category verification Conformity verification	Group A
Structural	Parallel analysis/exploratory structural equation	
	Confirmatory factor analysis	Group B
External	Latent mean analysis	

**Table 4 ijerph-19-03433-t004:** Taekwondo Athletes’ 10 Key Sport Psychological Skills.

Factor	Definition
Relaxation	Skill to relieve psychological tension caused by negative emotions through muscle tension.
Self-talk	Skill to overcome difficulties and encourage oneself through self-talk.
Emotion regulation	Skill to control the intensity of negative emotions or convert them into positive emotions.
Imagery	Imagining successful scenarios or important behavioral skills and movements.
Goal setting	Setting specific goals for training and competition.
Routine	Consistent behaviors or habits performed to maintain high athletic performance.
Confidence	Positive belief in one’s own abilities and performance.
Concentration	Skill to focus on training and competition regardless of circumstances.
Fighting spirit	Willingness and behavior to do one’s best until the end.
will	Firm commitment to achieve a goal or to win.

**Table 5 ijerph-19-03433-t005:** Items Assessing Taekwondo Athletes’ Sport Psychological Skills.

Item #	V Coefficient	Item #	V Coefficient	Item #	V Coefficient
4	0.13	21	0.09	43	0.15
7	0.18	25	0.06	46	0.07
13	0.11	30	0.14	48	0.14
14	0.03	36	0.03		

**Table 6 ijerph-19-03433-t006:** Verification of Unidimensionality.

		Eigenvalue	%
Explained variance	Person	21.4	25.8
	Item	13.6	16.4
Unexplained variance		48.0	57.8
Total variance		83.0	100

**Table 7 ijerph-19-03433-t007:** Verification of the 5-point Likert Scale’s Appropriateness.

Category	Count	%	AM	Infit	Outfit	SC	[SC]
1	301	3	−0.54	1.08	1.24	NONE	NONE
2	1022	9	0.02	0.97	0.98	−1.52	1.48
3	2988	26	0.61	1.03	1.02	−0.98	0.54
4	3449	32	1.16	0.99	0.98	0.59	1.57
5	3021	28	1.77	0.99	1.01	1.81	1.22

**Table 8 ijerph-19-03433-t008:** Verification of 3-point Likert Scale’s Adequacy.

Category	Count	%	AM	Infit	Outfit	SC	[SC]
1	1201	11	−1.07	1.00	1.24	NONE	NONE
2	7882	66	0.58	0.99	0.98	−1.52	1.48
3	3244	23	1.91	1.01	1.02	−0.98	0.54

**Table 9 ijerph-19-03433-t009:** Results of Item Relevance Verification.

Items	Logit	MNSQ		PBC	Items	Logit	MNSQ		PBC
		Infit	Outfit				Infit	Outfit	
3	0.82	1.49	1.51	0.45	33	−0.44	0.07	0.68	0.49
10	1.02	1.91	1.90	0.40	35	1.37	1.40	1.42	0.57
16	1.54	1.88	1.84	0.50	38	1.22	1.50	1.49	0.42
17	0.93	1.68	1.62	0.51	41	−0.47	0.69	0.71	0.54
23	−0.47	1.51	1.53	0.47	47	−0.38	0.72	0.74	0.59
28	−0.51	1.88	1.89	0.42	50	1.07	1.63	1.60	0.44

**Table 10 ijerph-19-03433-t010:** Parallel Analysis Results.

Factor #	Real-Data Eigenvalues	95% of Random Eigenvalues	Comparison
1	13.223	2.443	Real-data ˃ 95% random
2	2.544	2.044	Real-data ˃ 95% random
3	2.332	1.965	Real-data ˃ 95% random
4	1.993	1.803	Real-data ˃ 95% random
5	1.774	1.712	Real-data ˃ 95% random
6	1.593	1.632	Real-data ˂ 95% random

**Table 11 ijerph-19-03433-t011:** Comparison of Factors’ Fit.

Number of Factors	χ^2^	*df*	RMSEA	RMSEA (90% CI)
3	2343.11	834	0.089	(0.081, 0.089)
4	2132.22	804	0.076	(0.070, 0.080)
5	19,884.34	721	0.072	(0.067, 0.075)
6	1632.23	694	0.068	(0.061, 0.069)
7	1488.34	654	0.064	(0.059, 0.068)

Note. RMSEA: root-mean-square error of approximation.

**Table 12 ijerph-19-03433-t012:** Results of ESEM Analysis.

	Item			1	2	3	4	5	6
1	During training, I set goals and engage in practice			0.754 *	0.100	0.032	0.199	0.038	0.036
2	I set the training goals and methods based on the game results			0.920 *	0.043	−0.004	0.201	0.057	0.044
4	I train based on the daily, weekly, and monthly goals			0.602 *	0.018	0.021	0.122	0.122	0.087
8	When I play against an opponent, I don’t feel timid			0.109	0.803 *	0.122	0.015	0.104	−0.005
6	When I play against my rivals, I gain confidence			0.105	0.772 *	−00.36	−0.002	0.009	0.074
5	Every time I play, I believe in my own abilities			−0.044	0.564 *	−0.044	0.123	0.010	0.086
23	I often imagine highly technical scenes			−0.100	0.055	0.665 *	0.099	0.111	0.203
25	I always imagine a scene when a skill is succeeding			−0.049	0.211	0.702 *	0.098	0.211	0.211
22	I draw movement techniques according to the type of opponent			−0.010	0.204	0.800 *	0.122	0.106	0.005
34	I tell myself to be strong in every difficult moment			0.104	0.104	−0.199	0.596 *	0.104	0.223
31	When I lose a match, I encourage myself by talking to myself			0.009	0.009	−0.023	0.822 *	0.009	0.211
35	I always have frequent encouraging conversations with myself			0.010	0.010	−0.108	0.795 *	0.010	0.102
27	I promise to win every time I play			0.057	−0.012	0.078	0.050	0.455 *	0.076
26	I promise I will never lose in training and competition			0.044	−0.037	−0.060	0.100	0.596 *	−0.046
28	When playing against a strong athlete, the game is played with a focus on attack			0.031	−0.054	0.003	0.105	0.662 *	−0.008
41	Even in difficult situations, concentration is not disturbed			0.053	0.069	0.032	−0.022	−0.050	0.731 *
44	I don’t lose concentration when I make a mistake during a game			0.067	−0.044	0.038	0.029	0.015	0.716 *
42	Even wandering around during the game does not weaken my concentration			−0.020	0.068	0.018	0.093	0.046	0.539 *
	A	B	C	D		E		F	
B	0.439	1							
C	0.332	0.302	1						
D	0.324	0.340	0.362	1					
E	0.282	0.403	0.353	0.403		1			
F	0.402	0.400	0.400	0.332		0.355		1	

Note. χ2 = 998.43, df = 823, CFI = 0.910, RMSEA = 0.047. * 0.05.

**Table 13 ijerph-19-03433-t013:** Factor Structure Model Fit.

χ^2^	*df*	RMSEA (CI)	TLI
672.33	393	0.053 (0.049, 0.062)	0.910

Note. TLI: Tucker–Lewis index.

**Table 14 ijerph-19-03433-t014:** Results of ML CFA.

Factor		SC	S.E	CR			AVE
Goal setting	1	0.655	0.044	0.962			0.902
	2	0.782	0.045				
	4	0.665	0.052				
Confidence	8	0.772	0.043	0.970			0.910
	6	0.797	0.066				
	5	0.766	0.029				
Imagery	23	0.668	0.048	0.977			0.923
	25	0.711	0.065				
	22	0.683	00.56				
Self-talk	34	0.699	0.055	0.983			0.944
	31	0.733	0.046				
	35	0.765	0.039				
Fighting spirit	27	0.727	0.055	0.961			0.908
	26	0.699	0.050				
	28	0.705	0.037				
Concentration	41	0.741	0.052	0.971			0.936
	44	0.872	0.048				
	42	0.818	0.034				
	Goal setting	Confidence	Imagery		Self-talk	Fighting	Concentration
Confidence	0.402	1					
Imagery	0.383	0.356	1				
Self-talk	0.377	0.397	0.392		1		
Fighting	0.352	0.487	0.366		0.445	1	
Concentration	0.414	0.433	0.410		0.402	0.377	1

Note. SC: standardized coefficient, CR: conceptual reliability.

**Table 15 ijerph-19-03433-t015:** Response and Item Reliability.

	Factor	SEP	REL	Infit	Outfit
RR	Goal setting	4.03	0.91	0.99	0.98
	Confidence	3.94	0.90	0.97	0.98
	Imagery	3.97	0.91	0.99	1.00
	Self-talk	4.02	0.93	0.96	0.97
	Fighting spirit	4.10	0.94	1.01	1.00
	Concentration	3.89	0.95	1.00	1.00
IR	Goal setting	3.92	0.89	0.99	0.98
	Confidence	4.01	0.90	1.03	1.00
	Imagery	4.05	0.87	0.98	0.99
	Self-talk	3.95	0.94	1.00	1.02
	Fighting spirit	3.99	0.89	0.97	0.99
	Concentration	3.87	0.90	0.98	0.98

Note. RR: response reliability, IR: item reliability.

**Table 16 ijerph-19-03433-t016:** Results of the Verification of Measurement Equivalence by Gender.

	X^2^	∆X^2^	*df*	*p*	*∆df*	RMSEA
Unconstrained model	759.33		262	0.113		0.043
Factor coefficient same constraint	782.19	22.86	278	0.122	16	0.049
Covariance equal constraint	820.23	60.90	293	0.143	45	0.058
Factor coefficient/Factor coefficient same constraint	851.47	92.14	331	0.151	69	0.069
Factor coefficient/Covariance/Error variance	879.21	119.88	343	0.110	81	0.073

## Data Availability

The data presented in this study are available on request from the corresponding author. The data are not publicly available due to privacy issues.

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
