# Peer review of "Sport Psychological Skill Factors and Scale Development for Taekwondo Athletes"

_ijerph, 2022, doi:10.3390/ijerph19063433_

Round 1

Reviewer 1 Report

I would like to thank you for the opportunity to review this paper.

First of all I would like to congratulate the authors because I found it an original and very interesting manuscript. It has been undoubtedly one of the best validation articles I have been able to review so far.

The study aims to identify the sport psychological skills of Taekwondo athletes and to develop a scale to measure these skills.

I consider that an adequate literature review has been done and that the introduction perfectly fulfills its objective.

Regarding the method, I consider that a very exhaustive, methodical, structured and perfectly studied work has been done. The authors have made a great effort in defining the whole procedure.

The results show perfectly the statistics and tests performed and the conclusions are clear.

As an aspect of improvement, I would like to see an appendix with the final version of the scale that would facilitate access to the instrument for the rest of the scientific community to be able to replicate it or even adapt it for use in other athletes.

Translated with www.DeepL.com/Translator (free version)

Author Response

Thank you for your detailed review.

I will try my best for a better article in the future.

Reviewer 2 Report

The topic is interesting and this questionnaire might be useful for different goals both in practice and science.
The authors demonstrated mastery of psychometric procedures including IRT and various SEM methods. A fairly representative sample of respondents was used.
Nevertheless, I have a number of comments that I consider important and I would like the authors to take them into account.
1. And I think this is the most important thing when developing a questionnaire: you need to check the criterion validity in one form or another. For example, more successful athletes score higher on certain scales. For some reason, the authors compare the factor structures in men and women and show that there are no differences, but the purpose of this comparison is not clear.
2. I would like to understand what is the specificity of Taekwondo and why is this questionnaire about taekwondo? In the content of the items, I did not see any reference to Taekwondo, maybe this questionnaire is universal? Its value then, on the one hand, increases, and on the other hand, we still lose the specifics of the declared sport, which undoubtedly exists.
More specific remarks follow.
3. What is “closed sports”? (Page 1, paragraph 3). I thought it was an indoor sport, but apparently not.
4. Table 5 lists the 11 weakest items, and the comments to this table list 12 points (point 18 originated from somewhere additionally).
5. Section 3.1.2. “As shown in Table 6, the explained variance was 42.2%, so the criterion of 20% or more observed variance, which is the condition for unidimensionality was satisfied. "- Here I would like to have a reference where this cutoff of 20% is justified
6. Table 11. Chi-square for the 5-factor model to be checked. Somehow this value is knocked out of the general series.
7. In tables 11 and 13, you need to make sure that the models are described correctly, because of the descriptions that are presented there (18 items, respectively 3, 4, 5, ... factors) the indicated number of degrees of freedom (what can be calculated independently based on the description model) fails. If everything is correct there, maybe the author will write me the regression and correlation equations used, from where it will be clear how these degrees of freedom are calculated.
8. Tables 12 and 14 show two correlation matrices. I would like to understand what is the difference between them.
9. On page 9, after table 14, there is this sentence: “Because the latter exceeded the former, we concluded that there was strong evidence of the scale’s discriminant validity.” I would like an explanation because the conclusion is not quite obvious.
10. In table 16 in line 4 in the first column: Factor coefficient/Factor coefficient same constraint. Maybe you need Factor coefficient/ Covariance?

11. Usually a good measurement method needs to be checked for retest reliability. This was not done here, but at least it would be nice for the authors to note the need for retest reliability for the future.

Author Response

  1. However, despite uncovering the relationship between performance and sport psychological skills, these studies also have limitations in that they have not determined the mechanisms or created measures of sport psychological skills specifically reflecting Taekwondo. Therefore, we argue that it is necessary to perform a structural exploration of Taekwondo athletes’ sport psychological skills and to develop a scale to measure them. This will lead to a more comprehensive understanding of Taekwondo athletes’ performance, including ways to maximize it through skills training.
  2. Sport psychological skills have long been used in closed loop skill sports such as shooting and archery, but their use is gradually expanding to open loop skill sports such as skating and gymnastics(Porter, K. & Foster 1986).
  3. the explained variance was 42.2%, so the criterion of 20% or more observed variance, which is the condition for unidimensionality was satisfied(DeMars, 2010).

     DeMars, C. (2010). Item response theory. Oxford, England: Oxford University Press

  4. I deleted it as you said.

  5. Thank you for your words and I'll reflect it in my future research

Reviewer 3 Report

At the outset, I would like to thank you for the opportunity to review your work.
The topic has a significant contribution to the development of sports psychology.
The authors showed great diligence and methodological technique.
The only point that needs to be expanded is the discussion.
Relate the results to the personality of the fighter as a possible confounding variable.
I suggest to refer to some works
- Personality profile of combat sports champions against neo-gladiators. Arch Budo 2020; 16: 281-293
- Self-defence as a utilitarian factor in combat sports, modifying the personality of athletes at a champion level.
Arch Budo Sci Martial Art Extreme Sport 2020; 16: 71-77

Author Response

Thank you for your detailed review.

I refer to the article you suggested.

Thanks.
